# Novel Thymoquinone Nanoparticles Using Poly(ester amide) Based on L-Arginine-Targeting Pulmonary Drug Delivery

**DOI:** 10.3390/polym14061082

**Published:** 2022-03-08

**Authors:** Eman Zmaily Dahmash, Dalia Khalil Ali, Hamad S. Alyami, Hussien AbdulKarim, Mohammad H. Alyami, Alhassan H. Aodah

**Affiliations:** 1Department of Applied Pharmaceutical Sciences and Clinical Pharmacy, Faculty of Pharmacy, Isra University, Amman 11622, Jordan; hussein.abdalkarem@yahoo.com; 2Department of Physiotherapy, Faculty of Allied Medical Sciences, Isra University, Amman 11622, Jordan; dalia.ali@iu.edu.jo; 3Department of Pharmaceutics, Faculty of Pharmacy, Najran University, Najran 55461, Saudi Arabia; mhalmansour@nu.edu.sa; 4National Center of Biotechnology, Life Science & Environment Research Institute, King Abdulaziz City for Science and Technology, Riyadh 11442, Saudi Arabia; aaodah@kacst.edu.sa

**Keywords:** thymoquinone, poly(ester amide), respiratory drug delivery, COVID-19, L-arginine, nanoparticles

## Abstract

Thymoquinone (*TQ*), the main active constituent of *Nigella sativa*, has demonstrated broad-spectrum antimicrobial, antioxidant, and anti-inflammatory effects, which suggest its potential use in secondary infections caused by COVID-19. However, clinical deployment has been hindered due to its limited aqueous solubility and poor bioavailability. Therefore, a targeted delivery system to the lungs using nanotechnology is needed to overcome limitations encountered with *TQ*. In this project, a novel *TQ*-loaded poly(ester amide) based on L-arginine nanoparticles was prepared using the interfacial polycondensation method for a dry powder inhaler targeting delivery of *TQ* to the lungs. The nanoparticles were characterized by FTIR and NMR to confirm the structure. Transmission electron microscopy and Zetasizer results confirmed the particle diameter of 52 nm. The high-dose formulation showed the entrapment efficiency and loading capacity values of *TQ* to be 99.77% and 35.56%, respectively. An XRD study proved that *TQ* did not change its crystallinity, which was further confirmed by the DSC study. Optimized nanoparticles were evaluated for their in vitro aerodynamic performance, which demonstrated an effective delivery of 22.7–23.7% of the nominal dose into the lower parts of the lungs. The high drug-targeting potential and efficiency demonstrates the significant role of the *TQ* nanoparticles for potential application in COVID-19 and other respiratory conditions.

## 1. Introduction

COVID-19 has caused an emergent pandemic that has resulted in an extraordinary worldwide health crisis [1,2]. It is characterized by inflammation, oxidation, and an exaggerated immune response, which in some cases can progress into a cytokine storm, which may develop into acute respiratory distress syndrome (ARDs) [3]. Despite the presence of a few effective medications that have emerged to assist in the treatment of infected patients with COVID-19, there are some natural herbal options that may assist in the prevention and management of COVID-19 [3,4,5,6,7,8].

Thymoquinone is the main active component of *Nigella sativa* and has demonstrated several pharmacological effects, such as immunomodulation, anticancer, anti-inflammatory, antiasthmatic, and antioxidant effects [9,10,11,12,13]. Recently, research has been focused on its benefits for the prevention and cure of COVID-19 [4,6,7]. Research results have indicated that *TQ* can downregulate inflammatory cytokines, reduce the nitric oxide level, improve organ functions [8], and may inhibit SARS-CoV-2 and interfere with its binding to ACE2 receptors. This can prevent virus entry and replication inside the host cell [14,15]. Findings from a clinical trial [16] revealed that the use of *Nigella sativa* and honey on COVID-19 patients supported the viral clearance and reduced the seriousness of the illness [4,16]. Furthermore, studies demonstrated the antibacterial activity of *TQ* against several micro-organisms, including *Staphylococcus aureus* and *Staphylococcus epidermidis* [17]. Other researchers have reported on the broad-spectrum antimicrobial activity of *TQ*, which suggests its potential use in controlling secondary infections caused by COVID-19 [5].

Moreover, although *TQ* has promising potential for COVID-19 and other systematic and respiratory conditions, complete clinical deployment has been hindered due to its hydrophobicity, limited aqueous solubility (1.83 g/L), and poor bioavailability [18]. Consequently, a targeted delivery system is urgently needed to overcome the limitations encountered with *TQ*.

Compared to systemic drug delivery, inhalation treatment locally delivers the active pharmaceutical agent (API) to targeted sites within the respiratory system, thereby enhancing its efficacy and lowering its systemic side effects. Moreover, respiratory drug delivery offers several other advantages, such as avoiding first-pass metabolism and increasing patient compliance, being non-invasive [19]. The lungs are an attractive drug delivery route owing to their additional advantages, including the large surface area, thin epithelial layer, and high vascularization, which allows significant absorption of the inhaled dosage forms. Such properties allow for lower doses to be delivered, and hence reduced side effects [20,21].

Many strategies are employed to deliver APIs to the lungs for local or systemic effects, including the use of nanoparticles. Polymeric nanoparticles have attracted great attention as a potential drug delivery system to the lungs because they can efficiently deliver a wide range of therapeutic molecules to the targeted area within the respiratory system [22,23,24]. These fine particles can augment the rate and extent of dissolution of sparingly soluble drugs. Additionally, nanoparticles can produce systems with modified physicochemical properties. Combining the advantages of pulmonary drug delivery with the outstanding attributes of nanoparticles offers an approach with favourable properties [25].

To enable nanoparticles to reach the alveoli, they should be of a size that exceeds 0.5 µm. Therefore, nanoparticles are made as nanoaggregates that have an aerodynamic size range of 0.5 to 5 µm. Such particles will disintegrate once deposited in the designated areas within the respiratory system [25,26]. Several biodegradable polymers are used as carriers to develop nanostructures for pulmonary drug delivery, each with desirable advantages but undesirable drawbacks. However, the degradation process and the clearance of degradation products may affect the safety and biocompatibility of such polymers [27].

Among the different synthetic biodegradable polymers based on amino acids, L-arginine poly(ester amide)s (Arg-PEAs) have gained particular importance in the biomedical field due to the lower toxicity of their degraded products, as well their excellent mechanical and thermal properties. Furthermore, Arg-PEAs may improve cell membrane permeability and facilitate the absorption of large and small substances; this is due to the three nitrogen atoms present in their structure, allowing easy and safe penetration into the cells [28,29,30,31]. Synthetic biodegradable Arg-PEAs have various applications. For example, Arg-PEAs have been used for non-viral gene carriers due to their high cellular uptake nature and their cellular biocompatibility. Additionally, Arg-PEAs have shown minimal adverse effects on cell morphology, viability, or apoptosis, even if used in large doses with biocompatibility [32,33]. Another application of Arg-PEA is as a carrier for nucleic acid drugs. The condensing charge capability and the hydrophobicity of Arg-PEA allow it to bind with the nucleic acid drugs and make it a good candidate for improving drug delivery [34]. The Arg-PEA can also promote the transdermal delivery of insulin and enhanced diffusion and distribution of drugs through the skin [35].

To the best of our knowledge, this is the first study to investigate the delivery of the *TQ* nanoaggregate using poly(ester-amide) based on L-arginine. The present investigation aimed to synthesize a novel nanoparticulate delivery system for *TQ*, targeted to the lungs as a nanoaggregate to avoid first-pass metabolism and its distribution to a non-targeted site. This may lead to a decrease in peripheral side effects.

## 2. Materials and Methods

### 2.1. Materials

Thymoquinone (*TQ*) and diethylene glycol were obtained from Sigma (Pool, UK). The terephthaloyl chloride, *p*-toluenesulfonic acid monohydrate and L-arginine were obtained from Acros Organics (Geel, Belgium). The ethyl acetate, trifluoroacetic acid (TFA), dimethyl sulfoxide (DMSO), toluene, acetone, and acetonitrile were obtained from Alpha Chemika (Maharashtra, India). Glycerin was obtained from Labchem (Zelienople, PA, USA). Potassium hydroxide (KOH) pellets were obtained from Riedel-de Haën (Seelze, Germany). Methyl alcohol, diethyl ether, Tween-80, and HPLC-grade water were purchased from Tedia High Purity Solvents (Fairfield, CT, USA).

### 2.2. Monomer, Polymer, and Formula Synthesis

#### 2.2.1. Synthesis of di-*p*-Toluenesulfonic Acid Salt of O,O’-bis-(L-Arginine)-di-Ethylene Glycol Monomer

First, α-amino acid L-arginine (6.2 g, 40 mmol), diethylene glycol (1.2 g, 20 mmol), and *p*-toluenesulfonic acid monohydrate (7.0 g, 40 mmol) in 150 mL of toluene were added to a 250 mL round-bottomed flask equipped with a Dean–Stark trap and a condenser. Next, the reaction mixture was heated to reflux for 10 h with vigorous stirring until 2.2 mL (120 mmol) of water was evolved. Then, the reaction mixture was cooled to room temperature. The toluene was removed by decantation to leave a solid white product at the bottom of the reaction flask (the monomer). The formed monomer was filtered and washed extensively with acetone. The product was dried at 70 °C under vacuum for 8 h. The yield was 91%. FTIR (ν, cm^−1^): 3359 (NH), 1699 (C=O ester). ^1^H NMR (DMSO d6, 500 MHz): δ = 1.35–161 (m, 12 H, -CH_2_-CH_2_-CH_2_-NH of Arg), 2.25 (s, 6 H, CH_3_ of H_3_C-Ph-SO_3_-), 2.46 (s, 2 H, CH of Arg), 3.04,3.30 (t, 8 H, CH_2_CH_2_ of -(O)C–O-CH_2_-CH_2_-O), 7.10, 7.45 (d, 16 H, CH of H_3_C-Ph-SO_3_-), 7.59, 7.72, 8.71 (br signals, 16 H, NH of Arg). ^13^C NMR (DMSOd6, 125 MHz): δ = 21.2 (CH_3_ of H_3_C-Ph-SO_3_-), 28.5, 31.6, 40.7 (-CH_2_-CH_2_-CH_2_-NH of Arg), 53.9 (CH of Arg), 60.7, 72.7 (CH_2_CH_2_ of -(O)C–O-CH_2_-CH_2_-O), 128.7, 125.7 (CH of H_3_C-Ph-SO_3_-), 38.7, 145.2 (C of H_3_C-Ph-SO_3_-), 157.7 (C of-(NH) -(+NH_3_) C=NH-),172.8 (C=O ester). Anal. Cald. for C_44_H_66_N_8_O_17_S_4_: C, 47.73; H, 6.01; N, 10.12; S,11.58%. Found: C, 47.71; H, 6.08; N, 10.07; S, 11.49%. The splitting patterns were as follows: “s” for singlet, “d” for doublet, “t” for triplet, “q” for quartet, “m” for multiplet, and “br” for broad signal.

#### 2.2.2. Synthesis of Poly(ester amide) Based on L-Arginine

A mixture of di-*p*-toluenesulfonic acid salts of O,O’-bis-(L-arginine)-diethylene glycol (2.34 g, 2.5 mmol) and KOH (0.60 g, 5 mmol) was dissolved in distilled water (25 mL). The resulting solution was cooled in an ice water bath to 0–5 °C. Then, a terephthaloyl chloride solution (0.50 g, 2.5 mmol), which was dissolved in ethyl acetate (10 mL), was added dropwise to the aqueous solution under vigorous stirring. The reaction mixture was kept under stirring for 5 min at 0–5 °C and at room temperature for 10 min. Next, the precipitated formed polymer was filtered and washed several times with distilled water and diethyl ether. Finally, the last traces of the solvent were removed by freeze-drying the polymer for six hours. Yield: 76%. FTIR (ν, cm^−1^): 3296 (NH), 1791 (C=O ester), 1725 (C=O amide), ^1^H NMR (DMSO d6, 500 MHz): δ = 0.79–2.24 (m, 12 H, -CH_2_-CH_2_-CH_2_-NH of Arg), 2.23 (s, 6 H, CH_3_ of H_3_C-Ph-SO_3_-), 3.08, 3.46 (t, 8 H, CH_2_CH_2_ of -(O)C–O-CH_2_-CH_2_-O), 4.69 (s, 2 H, CH of Arg), 7.07, 7.45 (d, 8 H, CH of H_3_C-Ph-SO_3_-), 7.96 (CH of Ter), 8.07, 8.18, 8.75 (br signals, 10 H, NH of Arg), 8.23 (br signals, 2 H, NH of amide). ^13^C NMR (DMSO d6, 125 MHz): δ = 21.2 (CH_3_ of H_3_C-Ph-SO_3_-), 25.9, 28.4, 40.8 (-CH_2_-CH_2_-CH_2_-NH of Arg), 53.3 (CH of Arg), 47.3, 70.2 (CH_2_CH_2_ of -(O)C–O-CH_2_-CH_2_-O), 125.7, 128.7 (CH of H_3_C-Ph-SO_3_-), 129.7 (CH of Ter), 135.6, 136.7 (C of Ter), 138.5, 145.5 (C of H_3_C-Ph-SO_3_-), 157.5 (C of-(NH) -(+NH_3_) C=NH-), 166.1 (C=O amide), 167.7 (C=O ester).

#### 2.2.3. Synthesis of Thymoquinone Poly(ester amide) Nanoparticles

A mixture of di-*p*-toluenesulfonic acid salts of O,O’-bis-(L-arginine)-diethylene glycol (2.34 g, 2.5 mmol), and KOH (0.60 g, 5 mmol) was dissolved in distilled water (25 mL). The resulting solution was cooled in an ice water bath to 0–5 °C. Then, a terephthaloyl chloride solution (0.50 g, 2.5 mmol) and *TQ* were dissolved in ethyl acetate (10 mL). Three formulations were developed with increasing *TQ* amounts (50, 250, and 500 mg). The ethyl acetate solution was added dropwise to the aqueous solution under vigorous stirring. The reaction mixture was kept under stirring for 5 min at 0–5 °C and at room temperature for 10 min. Next, the precipitated polymer was filtered and washed several times with distilled water and diethyl ether. Finally, the last traces of the solvent were removed by freeze-drying the polymer for six hours.

### 2.3. Molecular Profiling and Characterization

Once the monomer, polymer, and *TQ*–polymer-loaded nanoparticles were synthesized, several profiling techniques were employed to enable the characterization of the products.

#### 2.3.1. Determination of Solution Viscosity and Molecular Weight

The viscosity of the polymer was measured using an Unbeholden glass capillary viscometer, Rheotek, Poulten Selfe and Lee Ltd. (Essex, UK) in a thermostated water bath at 25 ± 0.1 °C. The solution temperature was equilibrated for 10 min before conducting the viscosity measurements. The measurements were repeated until reproducible values were attained. The intrinsic viscosities were calculated from viscosity measurements of dilute polymer solutions (0.5 g /dL) in DMSO solvent. Solution viscosity was employed to estimate the polymer molecular weight. The molecular weight was calculated using the Mark-Houwink-Sakurada equation to give a relation between intrinsic viscosity and molecular weight, as can be obtained from the following Equation (1):(1)$$\left[\eta \right]=KM^{a}$$
where [*η*] is the intrinsic viscosity, “*M*” is the molecular weight, and “*K*” and “*a*” are constants that depend on the type of polymer and solvent. The constant “*a*” depends on the polymer–solvent interaction and temperature. Values of “*a*” are typically between 0.5 and 0.8. These values for our polymers were not available; the values of other polymers with similar solvents and temperatures were used to approximate the molecular mass [36].

#### 2.3.2. Fourier Transform Infrared (FTIR) Spectroscopy Analysis

The FTIR spectra of the monomer, the polymer, and the *TQ*–polymer-loaded nanoparticle samples were recorded in triplicate directly on the solid samples using a Perkin Elmer FTIR spectrometer (Akron, OH, USA) coupled with Spectrum 10 software, which was used to operate and treat the FTIR spectra. A sample of a few milligrams was loaded on the sample holder above a laser lens and held in place by screwing down the relevant adaptor. Each sample’s FTIR spectrum scans were obtained over the range of 4000–450 cm^−1^ with a resolution of 2 cm^−1^ by coadding 64 interferograms, with a measurement accuracy in the frequency data range at each measured point of 0.01 cm^−1^. The frequency of each band was obtained automatically by using the “find peaks” command of the instrument software.

Principal component analysis (PCA) of the FTIR spectra of *TQ*, the monomer, the polymer, and *TQ*-loaded formula were conducted using multivariate analysis with Minitab software according to the method described in [37,38]. The FTIR data sets of the obtained spectra were organized in matrices of 1101 × n measurable variables, where n is the number of samples (4). For each sample, the variables were the values of transmittance (%) associated with the wavenumbers (1101) in the range of 1800–700 cm^−1^. The FTIR spectra of all samples within the matrices were normalized in the range of 700 to 1800 cm^−1^ and then PCA analysis was carried out. Three principal components were used to explain the variations, whereby PC1 explained 86.2%, PC2 explained 7.25%, and PC3 explained 5.4% of the variations in transmission; therefore, a total of 99.1% of variations in transmission were explained. The loading and score plots were used to illustrate the main variations among samples.

#### 2.3.3. Nuclear Magnetic Resonance (NMR) Spectroscopy

The ^1^H-NMR (500 MHz) and ^13^C-NMR (125 MHz) of the monomer and polymer were recorded on a Bruker Avance DPX spectrometer, Bruker DPX-500, (Ettlingen, Germany) using tetramethylsilane (TMS) as the internal standard. All chemical shifts were given in parts per million (ppm) relative to the internal standard tetramethylsilane (TMS = 0.00 ppm). The splitting patterns were illustrated as follows: “s” for singlet, “d” for doublet, “t” for triplet, “q” for quartet, “m” for multiplet, and “br” for broad signal.

#### 2.3.4. Differential Scanning Calorimetry (DSC) Analysis

The DSC analysis of the *TQ* powder, the polymer, and the *TQ*–polymer-loaded nanoparticles was carried out using a DSC Q200-TA instrument by Mettler-Toledo, LLC (Columbus, OH, USA). A 2.5 mg sample was placed onto an aluminium pan and heated at a heating rate of 10 °C/min under continuous purging of nitrogen (50 mL/min).

#### 2.3.5. Particle Size, Polydispersity Index (PDI), and Zeta Potential Analysis

The average diameter, polydispersity index (PDI), and zeta potential of the polymer alone and the *TQ*–polymer nanoparticles were analyzed at 25 °C by photon correlation spectroscopy using a Zetasizer Nano ZS90 instrument from Malvern Instruments (Worcestershire, UK). Samples were diluted with deionized water before analysis and sonicated at medium amplitude (60 Hz) for 30 s. The analysis was performed in triplicate. Results are reported as mean ± standard deviation (SD).

#### 2.3.6. High-Performance Liquid Chromatography (HPLC) Assay Method for *TQ*

Quantification of *TQ* was conducted by employing a Dionex Softron HPLC system (Thermo Fisher Scientific Inc., Waltham, MA, USA) with a gradient pump and a UV detector set at 250 nm at 25 °C. A Fortis C18 column (4.6 mm × 250 mm, 5 μm) from Fortis Technologies Ltd. (Neston, UK) was used. The method was as follows: an isocratic method was followed for aerodynamic properties using a mobile phase of 80:20 (*v*/*v*) acetonitrile/water containing 0.125% (*v*/*v*) TFA and a flow rate of 1 mL/min for 12 min. Linearity was assessed between 3.9 µg/mL and 62.5 μg/mL using an injection volume of 20 μL. A stock solution of *TQ* in acetonitrile with a concentration of 1000 µg/mL was used to prepare the calibration curve. Serial dilutions were made using acetonitrile as the diluent. All samples were filtered using a membrane syringe filter with a 0.45 μm aperture size. The column was preconditioned for 30–60 min before use. All samples were made in triplicate. Results are expressed as means ± SD and relative standard deviation (RSD). The method was further validated according to the International Conference on Harmonization in terms of the specificity, accuracy, precision, limit of detection, and limit of quantification [39].

Quantification of *TQ* was conducted by employing a Dionex Softron HPLC system (Thermo Fisher Scientific Inc., Waltham, MA, USA) with a gradient pump and a UV detector set at 250 nm at 25 °C. A Fortis C18 column (4.6 mm × 250 mm, 5 μm) from Fortis Technologies Ltd. (Neston, UK) was used. The method was as follows: an isocratic method was followed for aerodynamic properties using a mobile phase of 80:20 (*v*/*v*) acetonitrile/water containing 0.125% (*v*/*v*) TFA and a flow rate of 1 mL/min for 12 min. Linearity was assessed between 3.9 µg/mL and 62.5 μg/mL using an injection volume of 20 μL. A stock solution of *TQ* in acetonitrile with a concentration of 1000 µg/mL was used to prepare the calibration curve. Serial dilutions were made using acetonitrile as the diluent. All samples were filtered using a membrane syringe filter with a 0.45 μm aperture size. The column was preconditioned for 30–60 min before use. All samples were made in triplicate. Results are expressed as means ± SD and relative standard deviation (RSD). The method was further validated according to the International Conference on Harmonization in terms of the specificity, accuracy, precision, limit of detection, and limit of quantification [39].

#### 2.3.7. Drug Loading Capacity

The amount of loaded *TQ* within the formulation was calculated according to two parameters: the entrapment efficiency (*EE*) and drug loading (*DL*), which were calculated according to the following formulas:(2)$$EE\;\left(\%\right)=\frac{TQ_{t\;}-TQ_{f}}{TQ_{t}}\times 100$$
(3)$$DL\;\left(\%\right)=\frac{TQ_{t}-TQ_{f}}{Formula_{t}}\times 100$$
where $TQ_{f}$ is the weight of free *TQ* in the supernatant and $TQ_{t}$ is the total weight of *TQ* used to prepare the nanoparticles, whereas $Formula_{t}$ is the total weight of the produced formula (polymer and drug). Briefly, the liquid that was produced from the synthesis and subsequent washing processes (supernatant) was accurately measured and the quantity of *TQ* was assessed using HPLC analysis.

#### 2.3.8. Transmission Electron Microscopy (TEM)

The morphology of the prepared *TQ*-loaded nanoparticles was determined using TEM. A drop of nanosuspension (using the high-dose formula) was placed on a paraffin sheet and a copper grid, which was placed on the sample and left for 1 min to allow the nanoparticles to adhere. The samples were further examined by TEM (Morgagni 268D; Fei Company, Hillsboro, OR, USA). ImageJ software was used for the calculation of the average particle size.

#### 2.3.9. X-ray Diffraction (XRD) Analysis

X-ray diffractometry (MiniFlex 600 benchtop diffractometer (RigaKu, Tokyo, Japan)) was used to investigate the physical form of the *TQ*, the poly(ester amide), and the drug-loaded polymer. The XRD experiments were performed over the 2θ range of 5 to 99°, with Cu Kα radiation (1.5148227 Å) at a voltage of 40 kV and a current of 15 mA. All samples were fixed on a glass holder and scanned in triplicate. The data was recorded at a scanning speed of 5°/min, and OriginPro^®^ software was employed to analyze the scans from OriginLab Corporation (Northampton, MA, USA).

#### 2.3.10. Elemental Analysis

The determination of the proportions of elements in polymers was conducted using a Euro-Vector 8910 Elemental Analyzer from Euro Vector Instruments and Software (Pavia, Italy).

### 2.4. Pulmonary Application of the Polymer–Thymoquinone-Loaded Nanoparticles

#### Aerodynamic Particle Size Analysis Using a Next-Generation Impactor (NGI)

Aerodynamic particle size distribution and in vitro deposition were established using an NGI (Copley Scientific Limited Model 170, Nottingham, UK) connected to a high-capacity pump (model HCP5, Copley Scientific, (Nottingham, UK) and a critical flow controller, model TPK 2000, Copley Scientific) (Nottingham, UK). A size 3 gelatine capsule (Pharmacare, Amman, Jordan) containing *TQ* formulation powder (approximately 20 mg each) was placed into an Aerolizer^®^ dry powder inhaler and dispersed into the NGI through the USP induction port at the flow rate of 60 L/min for 4 s per actuation to allow for 4 litres of air at a pressure drop of 4 kPa. To ensure accurate quantification of the *TQ*, 6 capsules per test were used. The NGI collection plates were coated with 1% *v/v* glycerine in acetone and allowed to dry for 30 min before use. After aerosolization, the deposited powder in each stage was dissolved with the acetonitrile and the *TQ* contents were analyzed by HPLC. Key aerodynamic parameters, namely the mass median aerodynamic diameter (MMAD), the geometric standard deviation (GSD), the emitted dose (*ED*), the fine particle fraction of the emitted dose (*FPF-E*), the fine particle fraction of the theoretical dose (*FPF-T*), and the respirable dose (*RD*), were calculated based on the dose deposited on the induction port (adapter and induction port), the preseparator, stages 1 to 7, and the micro-orifice collector (MOC). The formulas used for calculations of the aerodynamic parameters were as follows:(4)$$ED\;\left(\%\right)=\frac{\mathrm{Total}\;\mathrm{amount}\;\mathrm{of}\;\mathrm{TQ}\;\mathrm{from}\;\mathrm{induction}\;\mathrm{tube},\;\mathrm{preseprator}\;\mathrm{and}\;\mathrm{trays}\;1-7\;\&\;\mathrm{MOC}}{\mathrm{Total}\;\mathrm{amount}\;\mathrm{of}\;\mathrm{TQ}\;\mathrm{per}\;\mathrm{dose}}\times 100$$
(5)$$FPF-E\;\left(\%\right)=\frac{\mathrm{Total}\;\mathrm{amount}\;\mathrm{of}\;\mathrm{TQ}\;\mathrm{from}\;\mathrm{trays}\;2-7\;}{ED}\times 100$$
(6)$$FPF-T\;\left(\%\right)=\frac{\mathrm{Total}\;\mathrm{amount}\;\mathrm{of}\;\mathrm{TQ}\;\mathrm{from}\;\mathrm{trays}\;2-7}{\mathrm{Theoretical}\;\mathrm{dose}}\times 100$$
(7)RD (μg)=Total amount of TQ from trays 2−7

Regarding the MMAD and GSD, these were calculated based on the USP technique <601> [40]. All results were carried out in triplicate and reported as means ± SD.

### 2.5. Release Study and Release Kinetic Modelling

The in vitro release profile of *TQ* from the high-dose formula was assessed using a microplate spectrophotometer from Thermo Fisher (Vantaa, Finland). A total of 50 mg of the formula (made in triplicate) was placed in 2 mL of phosphate-buffered saline (PBS), at pH 6.8, while the wavelength was set at 250 nm. The device was set to record one reading every 60 min for 24 h. The cumulative amount of *TQ* released into the solution was measured at preset time intervals at the corresponding λ-max. The method was calibrated using PBS and a calibration curve was constructed over a range of 3.125–62.5 µg/mL. The concentration of the released *TQ* from *TQ*-loaded nanoparticles was expressed as a cumulative release percentage (%) of the total amount of *TQ* present in the nanoparticles using the calculated drug loading value. The cumulative releases of *TQ* as percentages were plotted against time.

The release pattern of *TQ* was fitted into five kinetic mathematical models, namely the zero-order release model, first-order release model, Higuchi model, Hixon-Crowell model, and Korsmeyer-Peppas model. The models are presented in the following equations (Equations (8)–(12)) [41]:

Zero-order model:(8)$$TQ_{t}=TQ_{0}+K_{0}\times t\;$$

First-order model:(9)$$Log\;TQ_{t}=Log\;TQ_{0}-\frac{K_{1}\times t}{2.303}$$

Higuchi model:(10)$$TQ_{t}=K_{H}\times \sqrt{t}.\;$$

Hixon-Crowell model:(11)$$\sqrt[{3}]{TQ_{rt}}=k\times t$$

Korsmeyer-Peppas model:(12)$$\frac{TQ_{t}}{TQ_{\infty }}=K\times t^{n}$$
where $TQ_{t}$ is the amount of *TQ* that is release at time *t*, $TQ_{0}$ is the initial amount of *TQ* (zero), $K_{0}$ is the zero-order rate constant, “*t*” is time, *K*_1_ is the first-order rate constant, *K*_*H*_ is the Highuchi rate constat, $TQ_{rt}$ is the reamining amount of *TQ*, “*k*” is the Keppa (a constant that is related to the geometry of the dosage form), $TQ_{\infty }$ is the total amount of *TQ* to be released, “*K*” is the release constant, and “n” is the release exponent (the value of “*n*” can be calculated by plotting the log fraction released versus log time—the slope of the line) [38,39,40].

### 2.6. Statistical Analysis

Data were analyzed with one-way analysis of variance (ANOVA) and Tukey’s post hoc test using Minitab version 18. All data were expressed as means ± standard deviation (SD) or relative standard deviation (RSD) and *p* < 0.05 was considered the level of significance.

## 3. Results and Discussion

### 3.1. Synthesis of O,O’-bis-(L-Arginine)-Diethylene Glycol

The condensation reaction between L-arginine and di-ethylene glycol in the presence of *p*-toluenesulfonate produced di-*p*-toluenesulfonic acid salts of O,O’-bis-(α-L-arginine)-diethylene glycol. The *p*-toluenesulfonic acid was added to protect the amino group against side reactions and esterification catalysts. Dean–Stark equipment was used in the condensation technique to eliminate the produce water (see Figure 1a).

### 3.2. Synthesis of Poly(ester amide)

The di-*p*-toluenesulfonic acid salts of O,O’-bis-(L-arginine)-diethylene glycol (diamine-diester) and terephthaloyl chloride were polymerized together using an interfacial polycondensation method. First, the diamine–diester monomer was dissolved in the aqueous layer with an inorganic base (KOH) and terephthaloyl chloride was dissolved in the ethyl acetate; at the interface between these two layers, the polymerization reaction took place. Then, KOH was used to deprotonate the protected amine group and neutralize the acid produced from the amine-acyl chloride reaction. The reaction required vigorous stirring to increase the available surface area, enable the formation of fine polymer particles, and reduce the encapsulation of solvents and reactants inside the polymer chain (see Figure 1b). Measuring the solution’s viscosity indicated the relative molecular weight of the polymer. The intrinsic viscosities of the polymers were calculated from the viscosity measurements of the dilute polymer solutions (0.5 g/dL) in DMSO at 25 °C. The polymer had intrinsic viscosity (0.32 dL/g). Therefore, the molecular weight was calculated using the Mark–Houwink–Sakurada equation (Equation (1)) [36]. Since the “K” and “a” values for our polymers were not available, the values of other polymers using the same solvent and temperature were used to approximate molecular mass calculation. The calculated molecular mass was 97,723 g/mol. This value indicated that the synthesized polymers had low to intermediate inherent viscosities, which implied that the produced polymer had low to moderate molecular mass.

### 3.3. Synthesis of Poly(ester amide) Thymoquinone Formula

Interfacial polycondensation could be used to prepare polymeric nanoparticles from a lipophilic monomer (terephthaloyl chloride) and hydrophilic monomer (O,O’-bis-(L-arginine)-diethylene glycol). The interface polycondensation was used for direct encapsulation of *TQ*, while the synthesis of the oily core polymer nanoparticles using this method resulted in particles with an average diameter of around 52 nm and a high-efficiency drug encapsulation amount of 99.77%. However, parts of *TQ* may be attracted to the surface of the polymer through the hydrogen bond (see Figure 2).

### 3.4. Molecular Profiling and Characterization

#### 3.4.1. FTIR Spectra Analysis

As depicted in Figure 3 and Table 1, FTIR spectra exhibited characteristic absorption bands for all organic functional groups of the monomer, the polymer, and the polymer–*TQ*-loaded nanoparticles. The di-*p*-toluenesulfonic acid salt of the O,O’-bis-(-L-arginine)- diethylene glycol monomer showed a stretching vibration band for the formed ester groups at 1699 cm^−1^. The FTIR spectra of the polymer and the *TQ*-loaded formula showed the same characteristic bands. The most important bands were the stretching vibrations for the formed amide carbonyl group at around 1725 cm^−1^ and the ester at around 1791 cm^−1^. Similar results were reported for FTIR functional groups based on poly(ester amide) from L-arginine [33,35].

From FTIR spectra of the four samples, it was possible to perform a multivariate analysis by conducting a PCA to identify certain similarities and differences between the samples. Figure 4A illustrates the scores plot in the plane of PC2 and PC3 for the samples, which together account for 99.1% of the total variability. Figure 4B shows the loadings plot using PC3 and PC2 versus the wavenumber, which indicate the absorption bands (wavenumber) mostly contributing to the variations among the samples for the PC2 and PC3 values.

Figure 4a revealed the variations in the spectra among samples, with the polymer and drug-loaded polymer (formula) shared similarities. This was expected, as both contained the same polymeric materials, and the difference was attributed to the presence of spectra related to *TQ*. The loading plot of PC2 vs. PC3 (Figure 4b) revealed variations between samples. Opposite contributions to the band are a clear indication of variations among samples. It is clear from Figure 4b that variations are obvious in the fingerprint range, which are a characteristic spectra for materials [41,42].

#### 3.4.2. NMR Spectra Analysis

The ^1^H-NMR and ^13^C-NMR characterized the synthesized monomer and polymer. The spectra exhibit characteristic chemical shifts corresponding to the different atoms present in the samples. The ^1^H-NMR spectrum of the synthesized di-*p*-toluenesulfonic acid salt of O,O’-bis-(-L-arginine)-diethylene glycol confirmed its proposed structure. A pair of doublets appeared at 7.08 and 7.24 ppm that were assigned to protons of the aromatic ring of the p-toluene sulfonic acid salt. The signal for the protons of the methyl group attached to the aromatic ring (Ar-CH_3_) appeared as a singlet at 2.25 ppm. The signals of diethylene glycol protons appeared as a triplet at 3.04 to 3.30 ppm. The protons of the carbons in the arginine were observed in the range of 1.35 to 2.46 as a multiplet. On the other hand, the ^1^H-NMR spectrum confirmed the poly(ester amide) structure based on the appearance of the protons of the terephthaloyl unit that appeared in the aromatic region at 7.96 ppm.

The structure of the monomer was also confirmed by ^13^C-NMR spectroscopy. The chemical shift of the formed ester carbonyl carbon was of special importance; it appeared at around 172.8 ppm. The two CH_2_ carbon peaks were observed at 60.7 and 72.7 ppm and were assigned to diethylene glycol carbons. The aromatic carbons of the *p*-toluenesulfonic acid salt were observed at 138.7, 128.7, 125.7, and 145.2 ppm. The carbons of the arginine side chain were observed at 53.9, 28.5, 31.6, and 40.7 ppm.

Further, the structure of the poly(ester-amide) was also confirmed by ^13^C-NMR. The chemical shift of the formed amide carbonyl carbon atom appeared at 166.1 ppm and the ester appeared at 167.7 ppm. The carbons of diethylene glycol were observed at 47.3 ppm and 70.2 ppm. The CH carbon signals of the aromatic rings of the terephthalate unit were observed at 129.7 ppm. A pair of peaks for the aromatic quaternary carbons of the same unit was observed in the chemical shift at 136.7 ppm. The ^1^H and ^13^C chemical shifts of the monomer and polymer are presented in Figure 5 and Table 2 [43,44,45,46].

#### 3.4.3. DSC Analysis

The DSC thermograms of *TQ*, the polymer, and *TQ*–polymer NPs, respectively, are shown in Figure 6. The results revealed a sharp and well-defined endothermic peak at 45.77 °C, representing the melting point of *TQ* (Figure 6a), followed by broad endothermic bands at 109 °C and 153 °C, corresponding to the decomposition process, and ending at 162.8 °C. Similarly, the DSC thermogram of the polymer (Figure 6b) showed an endothermic band that started at 151.53 °C and peaked at 162.92 °C, indicating the glass transition of the polymer. The additional broad endothermic band that ranged from 39 to 112 °C represented the dehydration process. The drug-loaded nanoparticle thermogram (Figure 6c) showed the characteristic peak of *TQ* at 45.62 °C and a broadened endothermic band at 163.81 °C. However, the broad band demonstrated an increase when compared with the polymer and an increase in enthalpy, which could be attributed to the encapsulation of the drug *TQ* in the polymer matrix [47,48].

#### 3.4.4. PSA, Zeta Potential, and TEM Analysis

The method used for the synthesis of the polymer, the interfacial polycondensation method, is known to produce nanoparticles. Table 3 summarizes the particle size parameters. It is noted that *TQ* as a powder demonstrated aggregated fine powder with a negative charge. Previous studies have also reported the negative charge of *TQ* [49]. Similarly, the formed polymer possessed a particle size of around 123.3 nm and a negative charge around 29 mV. Interestingly, the formed polymer—*TQ* nanoparticles presented a higher particle size and zeta potential but a better distribution. A higher PDI indicates the aggregated nature of the produced nanoparticles, which was further confirmed using TEM (Figure 5).

The TEM images as depicted in Figure 7 showed aggregated particles for the polymer and the polymer–*TQ* nanoparticles. Such aggregates were targeted, where efficient aerosolization and deposition of dry powder inhalers in the lowers parts of the lungs require the powder to be in the particle size range of 1–5 µm [21,26,47]. Once the particle has been deposited into the targeted surface, it will disintegrate into its primary nano-sized particles. This will provide ample surface area for the efficient release of the *TQ*. Despite the unclear interfaces between particles (see Figure 7), particle size analysis using ImageJ software showed that the polymer demonstrated an average particle size of 46.2 ± 12.47 nm, whereas for the *TQ*-loaded polymer the size was 51.4 ± 5.9 nm. The particle size assessed using the zetasizer was higher than that obtained from TEM. This could be attributed to the aggregation of the particles, as evident from the high PDI and low charge. However, the polymer is biodegradable, and upon deposition in the lower parts of the respiratory system it is expected to degrade slowly and release *TQ*.

#### 3.4.5. X-ray Diffraction Analysis

The XRD analysis of the *TQ* (Figure 8c) reveals the crystalline nature of *TQ*, with a characteristic sharp peak at 8.38 2θ. The polymer showed some degree of crystallinity, as evident from the presence of various peaks between 10 and 25 2θ (Figure 8b), accompanied by a heap indicating the semi-crystalline nature of the polymer, which was maintained upon the inclusion of the drug. Similar results showing the semi-crystalline nature of poly(ester amides) have been reported [50,51]. However, the *TQ* was not altered upon inclusion in the polymer (Figure 8a), indicating a physical and inclusion type of interaction. Such results could also be related to the charge of the drug and the polymer. Both materials possess negative charges with ionic interactions, and a physical inclusion is proposed between the polymer chains and/surface hydrogen bonding. Figure 8II illustrates the focused regions of *TQ* (Figure 8f), the polymer (Figure 8e), and the *TQ*-loaded nanoparticles (Figure 8d). The peaks showed that both the drug and polymer maintained their characteristic peaks (the rhombus shapes represent *TQ* and the circle shapes represent the *TQ*-loaded polymer).

#### 3.4.6. HPLC Method for Quantification of Thymoquinone

The HPLC method used for the *TQ* quantification was validated according to the ICH guidelines for analytical method validation [36]. The *TQ* peak was obtained at a retention time of 6.257 ± 0.056 min. There was no interference from the solvent front, which was eluted at 2.031 ± 0.066 min. To ensure the specificity of the method, the polymer was injected and it did not interfere with the peak of the *TQ*. A calibration curve was established by plotting the AUC against *TQ* concentrations that ranged from 3.9 to 62.5 µg\mL (see Figure 9). The regression equation was y=2.0213x+1.7792 (R2:0.9998).

The limit of detection (LOD) and limit of quantification (LOQ) of *TQ* were calculated using the standard deviation of the response and slope as specified by the ICH guidelines. The calculated LOD was 1.798 µg/mL and the LOQ was 5.447 µg/mL. To investigate the precision of the procedure, twelve samples of 31.25 µg/mL *TQ* solution were measured using the HPLC method, and the average was 100.97 ± 1.98 (RSD = 1.96%). The results indicated that the process is precise, as the RSD was below 2%. Inter- and intraday reproducibility and accuracy were assessed based on the recovery method, using 5 concentrations. The results in Table 4 show the good reproducibility and accuracy of the method.

#### 3.4.7. Pulmonary Application of Nanoparticles

Three formulations of *TQ* nanoparticles were developed, as can be seen in Table 5. The low *TQ* concentration formula (F1) showed low drug loading as the dose was low. Attempts were made to increase the *TQ* content. Increasing the *TQ* concentration resulted in higher drug loading of up to 35.56%, whereas F2 and F3 were assessed for their aerodynamic performance.

#### 3.4.8. In Vitro Assessment of Aerodynamic Performance

Two formulations were further used for the assessment of the in vitro performance using the NGI. The first formulation was a low-dose formula (F2) and the second was a large-dose formula (F3). The aerodynamic performance of the two formulations was promising. The results for the key inhalation parameters were assessed, namely the emitted dose (%ED), the FPF of the emitted dose (%FPF-E), the respirable dose (RD), and the FPF of the theoretical dose (FPF-T), as can be seen in Figure 10. The results revealed that the *TQ* nanoparticles were able to successfully produce emitted doses that exceeded 72.3% for F2 to 92.1% for F3. From Figure 10, F3 produced the highest amount of *TQ*, whereby all parameters (FPF-T, FPF-ED and the respirable dose) were significantly higher than F2 (one-way ANOVA, *p* < 0.05).

The percentage of the theoretical dose that was discharged from the capsules upon actuation was reported in the emitted dose. From Figure 10, the emitted doses were significantly different between F2 and F3 (*t*-test, *p* < 0.05). Despite the higher *TQ* content, the produced powder was light and fluffy, indicating the aerodynamic performance of the formulations.

An additional key variable was the FPF from the emitted dose, representing the percentage of the emitted dose that can reach the lower parts of the lungs and with an aerodynamic particle size of below 5 µm. Despite the higher percentage of the FPF-ED of F3 compared to F2, the difference was statistically insignificant (unpaired *t*-test, *p* = 0.0163). As the two formulations had different *TQ* contents, we could not use this variable for comparison. The main variable that provided an insight into the amount of *TQ* that reached the lower parts of the respiratory system was the respirable dose. It represents the mass of the *TQ* measured in microns that would reach the lower parts of the respiratory system. The F3 formulation showed the highest RD (474.8 µg) per actuation. The *t*-test analysis demonstrated that there was a significant difference among formulations (*p* = 0.010).

The mass median aerodynamic diameter (MMAD) values for the two formulations using the NGI set at 60 L/min are depicted in Table 6. The results revealed that the MMAD values for the aggregated nanoparticles ranged from 1.68 to 1.91 µm. However, there was a statistically significant difference among them (*t*-test, *p* = 0.0208), whereby increasing the *TQ* concentration resulted in a slightly larger particle size; these results did not affect the aerodynamic parameter formulation, as it was still within the targeted aerodynamic size range. Moreover, the geometric standard deviation (GSD) revealed that the two formulations produced particles with a similar particle size spread (GSD) (one-way ANOVA, *p* = 0.125). Such results were expected, as the method produces particles with a narrow particle size distribution. Additionally, the results demonstrated a successful technique to deliver *TQ*-loaded nanoparticle powder extract to the lungs, where the nanoparticles upon drying produced nanoaggregates within the particle size range of 1–5 microns, which enabled dispersion through the respiratory system. It is expected that upon deposition on the alveoli, the nanoaggregates will further disintegrate to produce smaller nanosized particles, hence avoiding quick clearance with possible local effects, as depicted in the TEM images (Figure 6).

The aerodynamic performance of the low-dose (F2) and high-dose (F3) formulations showed differences in the pattern of the particle size distribution (see Figure 11). Stages 2 to 7 are where a deposition is produced within the lower parts of the pulmonary system. The two formulations had a bi-modal aerodynamic particle size distribution within the extra-thoracic deposited particles, while the aerodynamic size exceeded 8 µm (particles deposited at the mouthpiece, induction tube, and stage 1). A good percentage was deposited within the particle size range of 1.66–4.46 µm (stages 2–5), indicating that the produced aerosol’s particle size was low where the smallest proportion was produced, in the range of 0.34 to 1.66 µm (stages 6–8). Although the percentage of produced F3 particles was higher, the trend of the aerodynamic particle size distribution was similar.

#### 3.4.9. Release Study and Release Kinetics

The *TQ* release from the nanoparticles, as can be seen from Figure 12, demonstrated an initial burst effect of *TQ* attracted to the surface of the polymer (around 50% released in the first 2 h). That was followed by a slower extended release over 22 h, representing the release of *TQ* that is incorporated within the polymer.

The release profile was modeled into the five main mathematical models that describe the kinetics of the release profile of *TQ*. Appendix A show the linear regression lines of each mathematical model. Table 7 lists the model constants and the coefficient of determination for each model. From the Table, the release profile fits the first-order release model, which represents the release of *TQ* from porous matrices [41]. However, a better understanding of the release pattern was obtained from the Korsmeyer-Peppas model, which showed the highest correlation. The release pattern is governed by the value of “n”. If “n” value was above 1, this indicates super case transport, which represents macromolecular relaxation of the polymeric chains [52].

## 4. Conclusions

In the present study, *TQ*-loaded poly(ester-amide) based on L-arginine nanoparticles were successfully prepared using the interfacial polycondensation method. The morphological and molecular profiling of the nanoparticles supported their suitability for pulmonary drug delivery, as the nanoparticles produce aggregates in a favourable micron size of 1–5 µm. The in vitro assessment of the aerodynamic particle size distribution revealed that almost 475 µg can be delivered to the lower parts of the respiratory system. The results suggest that pulmonary delivery of *TQ* could be a promising approach for optimal targeting, as well as for reducing systemic exposure. However, clinical studies need to be established scientifically to investigate its suitability in clinical practice for the management of COVID-19 and other respiratory conditions.

## Figures and Tables

**Figure 1 polymers-14-01082-f001:**
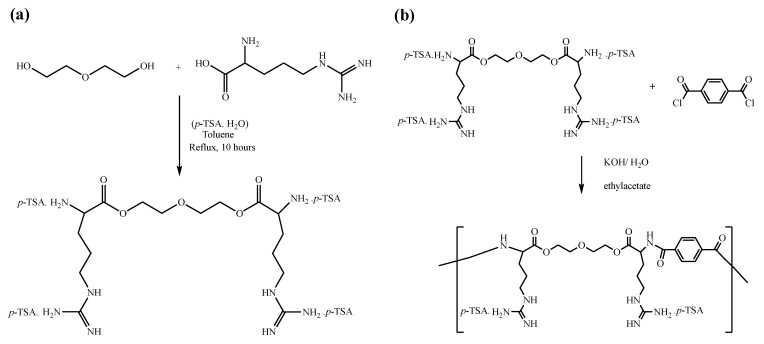
(**a**) Synthesis of di-*p*-toluenesulfonic acid salt of O,O’-bis-(L-arginine)-diethylene glycol monomer. (**b**) Synthesis of poly(ester amide)s by interfacial polycondensation of di-*p*-toluenesulfonic acid salt of O,O’-bis-(L-arginine)- diethylene glycol monomer.

**Figure 2 polymers-14-01082-f002:**
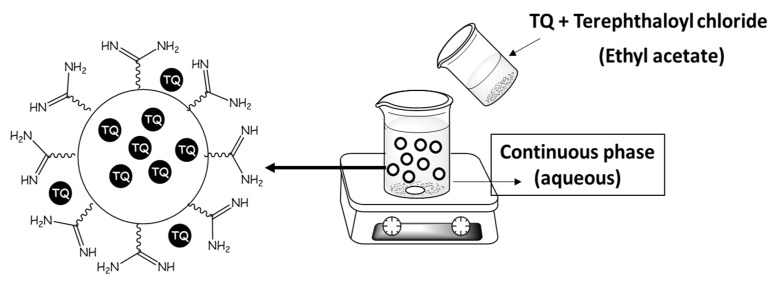
Schematic representation of interfacial polycondensation process and the formed *TQ*-loaded poly(ester amide) based on L-arginine, where part of the *TQ* is dispersed within the polymer and the other part is attracted to the surface of the polymer.

**Figure 3 polymers-14-01082-f003:**
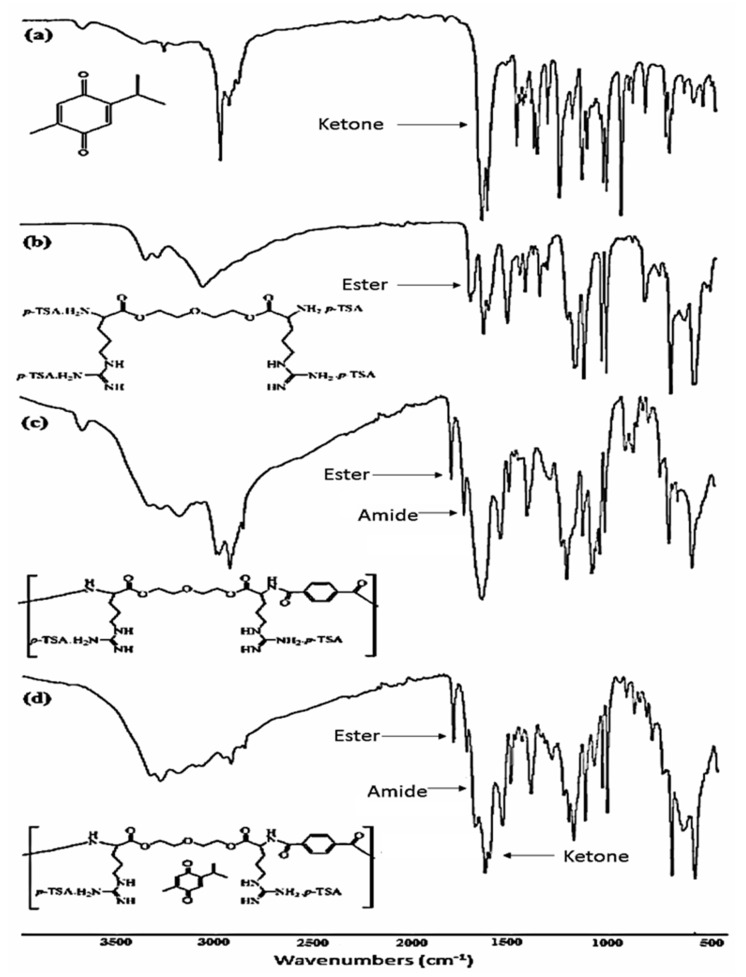
FTIR spectra: (**a**) *TQ*; (**b**) di-*p*-toluenesulfonic acid salt of O,O’-bis-(L-arginine)-diethylene glycol monomer; (**c**) poly (ester amide) based on L-arginine; (**d**) *TQ* poly(ester amide) formula.

**Figure 4 polymers-14-01082-f004:**
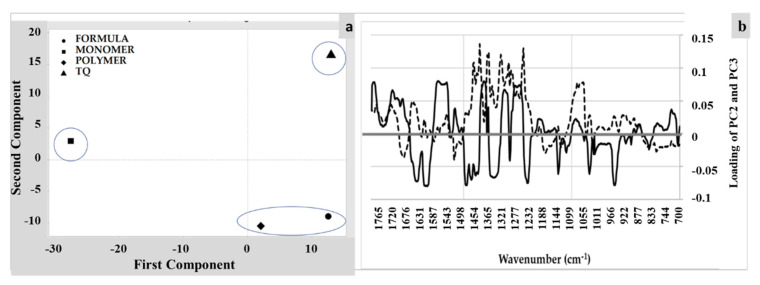
PCA of FTIR spectra (1800–700 cm^−1^) of *TQ*, the monomer, the polymer, and *TQ*-loaded polymer (formula): (**a**) score plot of PC1 (86.2%) vs. PC2 (7.5%) and (**b**) loading profiles of PC2 and PC3 within the allocated range (the dotted line is the loading profile of PC3, while the solid black line represents the loading profile of PC2).

**Figure 5 polymers-14-01082-f005:**
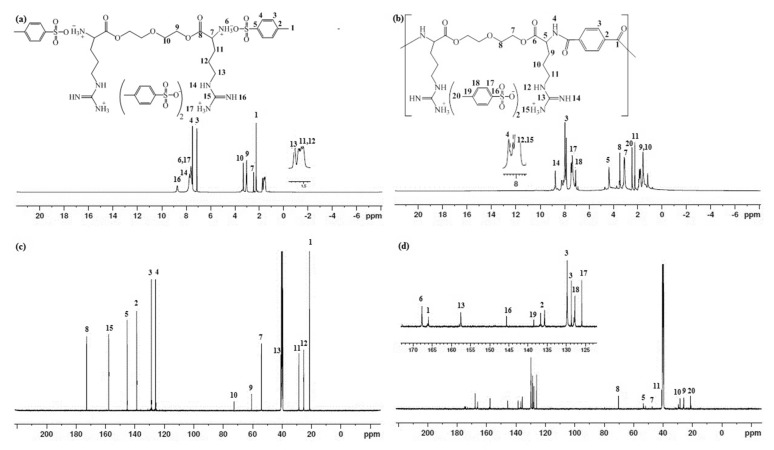
(**a**) ^1^H NMR of di-*p*-toluenesulfonic acid salt of O,O’-bis-(L-arginine)-diethylene glycol monomer (experiments were recorded at 500 MHz). (**b**) ^1^H NMR of poly(ester amide) based on L-arginine. (**c**) ^13^C NMR of di-*p*-toluenesulfonic acid salt of O,O’-bis-(L-arginine)-diethylene glycol monomer. (**d**) ^13^C NMR poly (ester amide) based on L-arginine. Experiments were recorded at 125 MHz.

**Figure 6 polymers-14-01082-f006:**
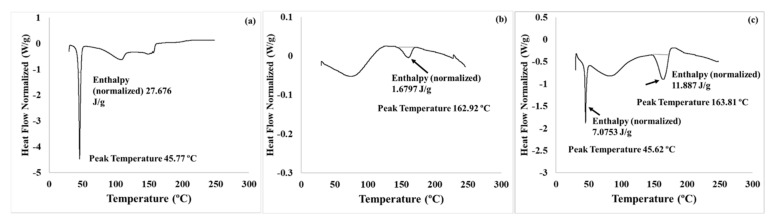
Differential scanning calorimetry (DSC) thermograms: (**a**) *TQ* powder; (**b**) the polymer; (**c**) *TQ*–polymer nanoparticles highlighting the enthalpy and melting points.

**Figure 7 polymers-14-01082-f007:**
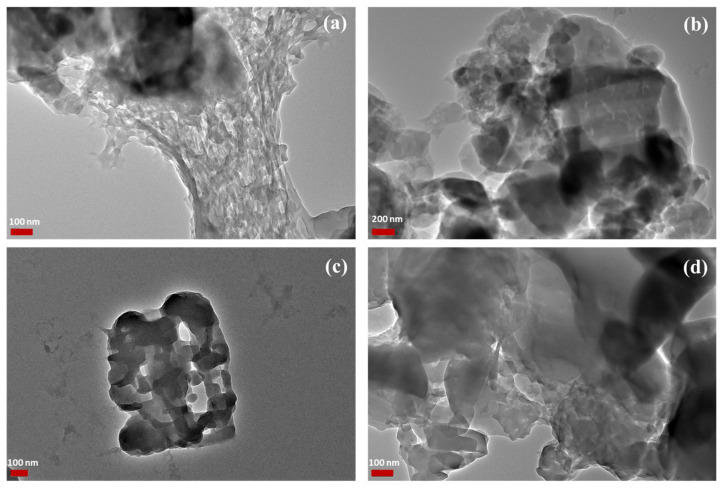
TEM images of (**a**,**b**) the poly (ester amide) based on L-arginine and (**c**,**d**) *TQ*-loaded nanoparticles. Scales: (**a**) 100 nm, (**b**) 200 nm, (**c**) 100 nm, and (**d**) 100nm.

**Figure 8 polymers-14-01082-f008:**
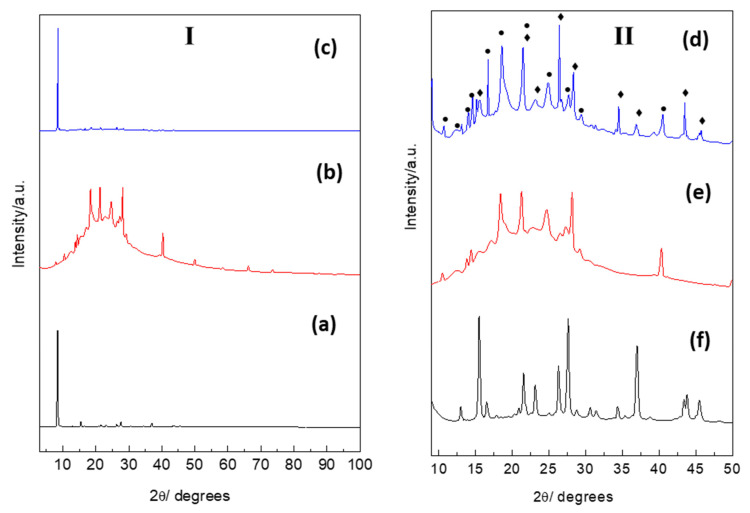
XRD patterns for panel [**I**]: (**a**) *TQ* powder; (**b**) polymer; (**c**) *TQ*–polymer nanoparticles. Panel [**II**] focus section from 10 to 50°: (**d**) *TQ* powder; (**e**) polymer; (**f**) *TQ*–polymer nanoparticles. Note in (**II**-**d**), the rhombus shapes over the peaks are related to *TQ*, while the circles are for the *TQ*-loaded polymer.

**Figure 9 polymers-14-01082-f009:**
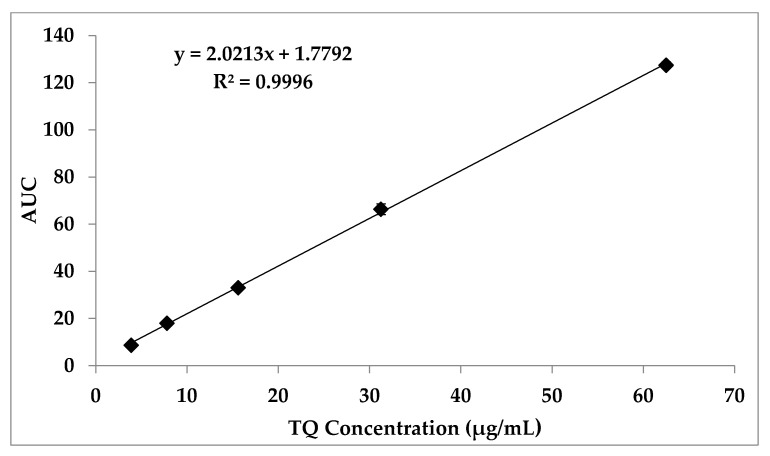
Calibration curve of *TQ* over a concentration range of 3.9 to 62.5 µg/mL.

**Figure 10 polymers-14-01082-f010:**
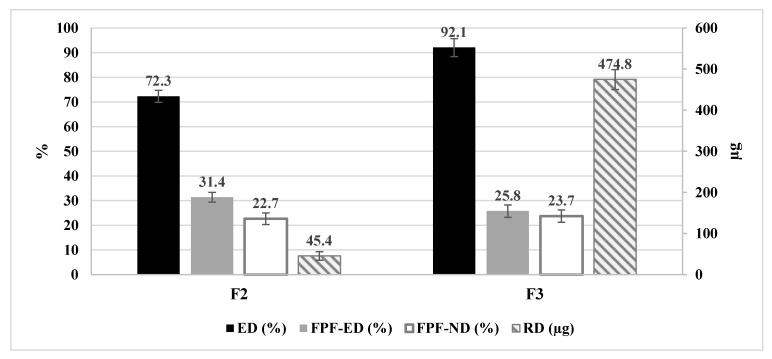
Aerodynamic performance of *TQ*-containing nanoparticles, highlighting the main parameters for formulations F2 and F3. ED: emitted dose; FPF-ED: fine particle fraction of emitted dose; FPF-ND: fine particle fraction of nominated dose; RD: respirable dose. Results are presented as means ± SD, *n* = 3.

**Figure 11 polymers-14-01082-f011:**
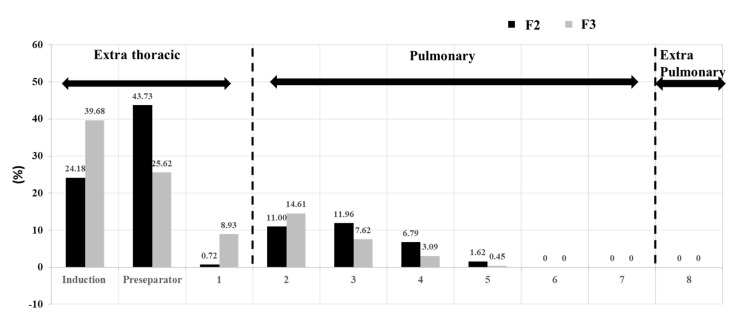
Aerodynamic particle size distribution of *TQ*-containing nanoparticles, highlighting the percentage of particles deposited the pulmonary section, which is the lower part of the respiratory system. Results are presented as means ± SD, *n* = 3.

**Figure 12 polymers-14-01082-f012:**
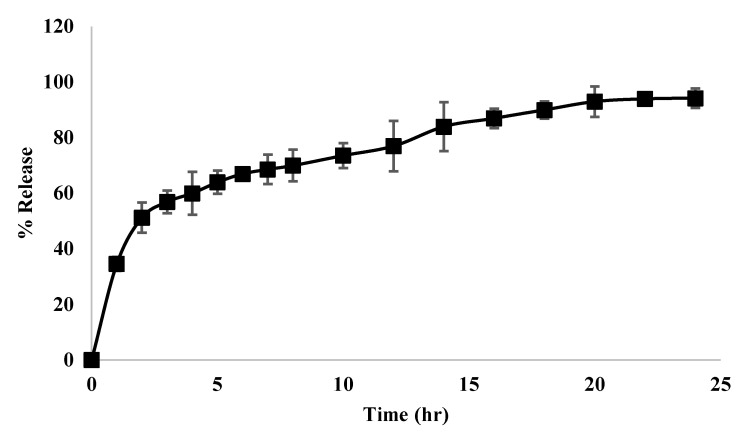
Release profile of *TQ* from *TQ*-loaded poly(ester amide) based on L-arginine nanoparticles (F3) for 24 h (means ± SD, *n* = 3).

**Table 1 polymers-14-01082-t001:** FTIR spectral properties of the monomer, polymer, and *TQ*–polymer.

	CH (cm^−1^) Aromatic	CH (cm^−1^) Aliphatic	NH (cm^−1^) Stretch	NH (cm^−1^) Bend	-COOR (cm^−1^)	-CON (cm^−1^)
Monomer	3014	2912	3359	1620	1699	-
Polymer	2995	2912	3296	1632	1791	1725
*TQ*–polymer	2946	2913	3279	1631	1791	1726

**Table 2 polymers-14-01082-t002:** ^1^H NMR and ^13^C NMR data of the synthesized monomer and polymer.

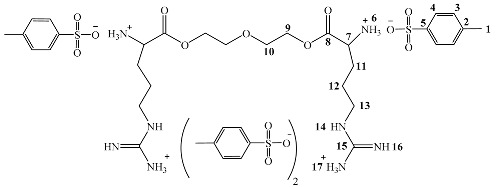																				
Chemical shift (ppm)																				
	1	2	3	4	5	6	7	8	9	10	11	12	13	14	15	16	17			
^1^H	2.25	-	7.10	7.45	-	7.59	2.46	-	3.04	3.30	1.35	1.58	1.61	7.72	-	8.71	7.59			
^13^C	21.2	138.7	128.7	125.7	145.2	-	53.9	172.8	60.7	72.7	28.5	31.6	40.7	-	157.7	-	-			
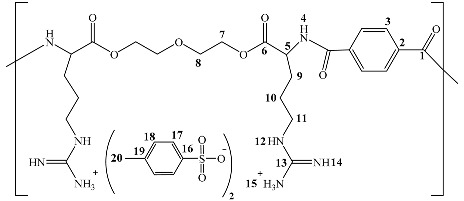																				
Chemical shift (ppm)																				
	1	2	3	4	5	6	7	8	9	10	11	12	13	14	15	16	17	18	19	20
^1^H	-	-	7.96	8.23	4.69	-	3.08	3.46	0.79	1.93	2.24	8.18	-	8.75	8.07	-	7.45	7.07	-	2.23
^13^C	166.1	136.7 135.6	129.7	-	53.3	167.7	47.3	70.2	28.4	25.9	40.8	-	157.5	-	-	145.5	125.9	128.6	138.5	21.2

**Table 3 polymers-14-01082-t003:** Particle size parameters of *TQ*, the polymer, and *TQ*–polymer-loaded nanoparticles (means ± SD, *n* = 3).

	Material	Thymoquinone	Polymer	Polymer *TQ*-Loaded NP
Parameter				
Particle size (nm)		390.1 ± 20.43	123.3 ± 11.65	187.41 ± 18.32
Polydispersity index (Pdi)		0.751 ± 0.12	0.451 ± 0.121	0.382 ± 0.180
Zeta potential (mV)		−42.54 ± 4.65	-29.59 ± 5.12	−19.25 ± 4.58

**Table 4 polymers-14-01082-t004:** Precision and reproducibility of *TQ* analysis and HPLC method for assessing inter- and intraday reproducibility.

Thymoquinone Concentration (µg/mL)	62.5	31.75	15.625	7.8125	3.90625
Intraday % Recovery (mean ± SD) (*n* = 3)	98.04 ± 1.64	99.49 ± 1.09	102.28 ± 2.01	99.03 ± 1.68	102.75 ± 0.98
Interday % Recovery (mean ± SD) (*n* = 9)	96.45 ± 1.98	100.07 ± 1.87	101.56 ± 2.04	100.76 ± 1.88	101.98 ±1.23

**Table 5 polymers-14-01082-t005:** *TQ*-loaded poly(ester amide) based on L-arginine nanoparticles, showing the composition, EE%, and DL% data.

	Formula	F1	F2	F3
Parameter				
Monomer (mg)		2000	1000	1000
Thymoquinone (mg)		50	250	500
*EE (%)*		89.82	92.13	99.77
*DL (%)*		7.45	22.80	35.56

**Table 6 polymers-14-01082-t006:** MMAD and GSD results for the two *TQ*-loaded poly(ester amide)s based on L-arginine nanoparticles.

	Formula	F2	F3
Parameter			
MMAD (µm)		1.679 ± 0.087	1.911 ± 0.065
GSD		0.086	0.091

**Table 7 polymers-14-01082-t007:** The constant and coefficient of determination (R^2^) and the “n” value for each mathematical model.

Mathematical Model	R^2^	Constant	n
Zero-order	0.823	3.71	-
First-order	0.982	7.03	-
Higuchi model	0.966	13.94	-
Hixon Crowell model	0.889	0.104	-
Korsmeyer–Peppas model	0.985	8.39	2.35

## Data Availability

All data related to this work is presented in the manuscript.

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
