# Peer review of "Novel Thymoquinone Nanoparticles Using Poly(ester amide) Based on L-Arginine-Targeting Pulmonary Drug Delivery"

_polymers, 2022, doi:10.3390/polym14061082_

Round 1

Reviewer 1 Report

On request of Polymers, I have revised the manuscript titled “Novel thymoquinone nano-encapsulated particles using poly(ester amide) based on L-arginine targeting pulmonary drug delivery for potential use in COVID-19” by Eman Zmaily Dahmasha et al.

The aim of the project reported in the present manuscript was to address the pharmacokinetic and pharmacodynamic issues of thymoquinone (TQ), to develop an antimicrobial, antioxidant, and anti-inflammatory device also gifted with immunomodulation effects, capable to target lungs and potentially usable in controlling secondary infections caused by COVID-19.

To this end, they successfully encapsulated TQ in a novel poly(ester-amide) based on L-arginine. The TQ-loaded nanoparticles (NPs) were characterized by several techniques, while the drug loading (DL%) and encapsulation efficiency (EE%) were determined by HPLC. The release profile of TQ from NPs was also investigated as well as the in vitro aerodynamic performance, which proved a delivery of 22.7-23.7% of the nominal dose into the lower parts of the lungs.

General Comments

Considering the historical period which we are living in from the year 2019, a study reporting on the development of a drug-loaded nanotechnological device with high drug-targeting potential and efficiency for potential application in COVID-19 and other respiratory conditions, is welcome. In this regard, this study can be considered of significant relevance and could attract the attention and interest of the Polymers audience, if well done. On the contrary, the study contains a lot of inaccuracies or even errors both of form and of concepts and contents that hamper the publication of this manuscript and must be addressed.

  • According to Polymers instructions, the abstract should be of 200 words at max. The abstract of the present manuscript does not respect this rule, being of 276 words. Please, shorten it accordingly.
  • The authors refer to the TQ-loaded nanomaterial as “nanocapsules” and it is incorrect. For definition, nanocapsules are constituted by an external polymeric membrane and an inner part composed of a liquid or polymeric matrix that contains the bioactive compound. According to what reported by the same authors, TQ is not only contained within the polymers, but also absorbed onto the surface. Accordingly, the TQ-loaded nanomaterials here reported are not nanocapsule, but more generally nanoparticles or nanodispersions.
  • The numbering of sections and subsections is dramatically wrong. In materials and methods there are only sections 1.1. and 1.1.1. and all the subsequent sections are numbered only using the number one‼‼
  • The template of Polymers is not respected, mainly in the format of the equations and Tables, in the way to indicate the Figures in the text and in the Figure captions (Figure 1a and not Fig.1(a)), in the way to report the references in the reference list, and in many other things. Please, check all manuscript and correct accordingly.
  • Some time, incorrectly, there are part in bold (lines 154-156, 186-195)!
  • The materials and method section needs to be improved. In the three syntheses, the authors should provide the peaks lists of the FTIR, 1H and 13C NMR analyses, opportunely reported, with indicated assignation. The signals multiplicity in the 1H NMR spectra peaks lists should be indicated using “s” for singlets, “d” for doublets and so on.
  • The authors should perform elemental analysis of the monomer or acquire its mass spectrum, while they should determine the MW of polymers by proper techniques.
  • The authors should provide more data concerning all the experiments to make them reproducible by other scientists.
  • Concerning equations 1 and 2. Equation 1 is correct, and it provide EE% even if the authors have not explained Dt. I suppose Dt means the total TQ used in the reaction. Please, correct. Equation 2 is correct, but it provides the drug loading (DL%) and not the encapsulation efficiency which has been provided by equation 1. Please, correct and correct also in the related Table 4.
  • Dots are necessary at the end of all Figure captions and Tables titles.
  • Figure 2 and Figure 3 should be improved: They are blurred.
  • To get more reliable information from FTIR spectra, the authors should process the matrix made of all spectral data using the principal components analysis (PCA), widely used for interpreting FTIR data.
  • In Figure 4 caption the authors should specify if spectra are acquired on the 300 MHz or 500 MHz instrument. Additionally, they should provide the MHz values of the acquisition the carbon spectra.
  • All tables should be adapted to the template and improved.
  • In all panels of Figure 6 the scale is invisible.
  • The authors should provide more data and information concerning the calibration model and should insert in the manuscript the related graph.
  • Concerning the release profile, the authors should determine the kinetic and the mechanisms which govern the TQ release by fitting the release data with four or five among the most used mathematical models. A detailed discussion of the results should be inserted.
  • It is not clear if the calibration model used for determining the drug loading and that used for the releases are the same or are different. In my opinion should be the same but the intervals of concentrations reported by the authors differ. Please, see 3.125-62.5 µg/mL versus 3.9-62.5 µg/mL.
  • Lines 510-522. Please adapt to the template.

I suggest Polymers to publish this manuscript only when the abovementioned issues will be addressed. Anyway, I decided to ask for major revisions.

On request of Polymers, I have revised the manuscript titled “Novel thymoquinone nano-encapsulated particles using poly(ester amide) based on L-arginine targeting pulmonary drug delivery for potential use in COVID-19” by Eman Zmaily Dahmasha et al.

The aim of the project reported in the present manuscript was to address the pharmacokinetic and pharmacodynamic issues of thymoquinone (TQ), to develop an antimicrobial, antioxidant, and anti-inflammatory device also gifted with immunomodulation effects, capable to target lungs and potentially usable in controlling secondary infections caused by COVID-19.

To this end, they successfully encapsulated TQ in a novel poly(ester-amide) based on L-arginine. The TQ-loaded nanoparticles (NPs) were characterized by several techniques, while the drug loading (DL%) and encapsulation efficiency (EE%) were determined by HPLC. The release profile of TQ from NPs was also investigated as well as the in vitro aerodynamic performance, which proved a delivery of 22.7-23.7% of the nominal dose into the lower parts of the lungs.

General Comments

Considering the historical period which we are living in from the year 2019, a study reporting on the development of a drug-loaded nanotechnological device with high drug-targeting potential and efficiency for potential application in COVID-19 and other respiratory conditions, is welcome. In this regard, this study can be considered of significant relevance and could attract the attention and interest of the Polymers audience, if well done. On the contrary, the study contains a lot of inaccuracies or even errors both of form and of concepts and contents that hamper the publication of this manuscript and must be addressed.

  • According to Polymers instructions, the abstract should be of 200 words at max. The abstract of the present manuscript does not respect this rule, being of 276 words. Please, shorten it accordingly.
  • The authors refer to the TQ-loaded nanomaterial as “nanocapsules” and it is incorrect. For definition, nanocapsules are constituted by an external polymeric membrane and an inner part composed of a liquid or polymeric matrix that contains the bioactive compound. According to what reported by the same authors, TQ is not only contained within the polymers, but also absorbed onto the surface. Accordingly, the TQ-loaded nanomaterials here reported are not nanocapsule, but more generally nanoparticles or nanodispersions.
  • The numbering of sections and subsections is dramatically wrong. In materials and methods there are only sections 1.1. and 1.1.1. and all the subsequent sections are numbered only using the number one‼‼
  • The template of Polymers is not respected, mainly in the format of the equations and Tables, in the way to indicate the Figures in the text and in the Figure captions (Figure 1a and not Fig.1(a)), in the way to report the references in the reference list, and in many other things. Please, check all manuscript and correct accordingly.
  • Some time, incorrectly, there are part in bold (lines 154-156, 186-195)!
  • The materials and method section needs to be improved. In the three syntheses, the authors should provide the peaks lists of the FTIR, 1H and 13C NMR analyses, opportunely reported, with indicated assignation. The signals multiplicity in the 1H NMR spectra peaks lists should be indicated using “s” for singlets, “d” for doublets and so on.
  • The authors should perform elemental analysis of the monomer or acquire its mass spectrum, while they should determine the MW of polymers by proper techniques.
  • The authors should provide more data concerning all the experiments to make them reproducible by other scientists.
  • Concerning equations 1 and 2. Equation 1 is correct, and it provide EE% even if the authors have not explained Dt. I suppose Dt means the total TQ used in the reaction. Please, correct. Equation 2 is correct, but it provides the drug loading (DL%) and not the encapsulation efficiency which has been provided by equation 1. Please, correct and correct also in the related Table 4.
  • Dots are necessary at the end of all Figure captions and Tables titles.
  • Figure 2 and Figure 3 should be improved: They are blurred.
  • To get more reliable information from FTIR spectra, the authors should process the matrix made of all spectral data using the principal components analysis (PCA), widely used for interpreting FTIR data.
  • In Figure 4 caption the authors should specify if spectra are acquired on the 300 MHz or 500 MHz instrument. Additionally, they should provide the MHz values of the acquisition the carbon spectra.
  • All tables should be adapted to the template and improved.
  • In all panels of Figure 6 the scale is invisible.
  • The authors should provide more data and information concerning the calibration model and should insert in the manuscript the related graph.
  • Concerning the release profile, the authors should determine the kinetic and the mechanisms which govern the TQ release by fitting the release data with four or five among the most used mathematical models. A detailed discussion of the results should be inserted.
  • It is not clear if the calibration model used for determining the drug loading and that used for the releases are the same or are different. In my opinion should be the same but the intervals of concentrations reported by the authors differ. Please, see 3.125-62.5 µg/mL versus 3.9-62.5 µg/mL.
  • Lines 510-522. Please adapt to the template.

I suggest Polymers to publish this manuscript only when the abovementioned issues will be addressed. Anyway, I decided to ask for major revisions.

Author Response

Dear Reviewer 

We would like to thank you for the valuable comments that will enrich the clarity of the manuscript. I attached a document that addressed all your comments. 

Reviewer 2 Report

The manuscript titled “Novel thymoquinone nanoencapsulated particles using poly(ester amide) based on L-arginine targeting pulmonary drug delivery for potential use in COVID-19” describes the loading thymoquinone onto the polymer. It possesses some merits for potential applications and can be considered to publish in Polymers after the corrections. The authors should follow the below comments:

  1. The discussion of FTIR and NMR spectra should cite previous reports. Moreover, the spectrum of TQ-polymer must be provided to conclude the presence of TQ in the drug delivery system.
  2. For TEM images, it seems that the samples did not exist in nano-scale. The interface between the particles is unclear. It seems that the particles were aggregated. The authors should correct the discussion section. This point is important to determine the effectiveness of drug delivery system.
  3. In the title, the authors have mentioned about the potential application of Covid-19 treatment. Thus, the authors must provide tests on the Virus inhibition.
  4. For release study, the authors should investigate release mechanism via study on kinetics theory and experiments (e.g. TEM images and XRD data of the systems after the release).
  5. Numbers of the heading should be corrected.

Author Response

Dear Reviewer 

We would like to thank you for your valuable comments, which will improve the clarity of the manuscript. We attached a document that addressed all the raised comments. 

Round 2

Reviewer 1 Report

Dear authors,

I have reconsidered the revised version of your manuscript noting that many of my requests have been met. However, much remains to be improved to make this manuscript publishable as an article on Polymers.

Regarding the formatting, even if the authors, I think in good faith, assert that they have adapted it to the Polymers template, in my opinion still many things do not respect it.

1) The titles of the sections and subsections do not respect this: the first letters of each word must be capitalized.
2) The formatting of the equations is incorrect.
3) The formatting of the part before the references is incorrect.
4) The formatting of the reference list is not correct, it does not respect the template.

Furthermore, Tables 1 and 3 must be improved by placing the data well in columns (Table 1) and making means and SDs stay on the same row (Table 3).

If (a), (b) etc. are written in the captions of the Figures, Figure 1a and not 1A must be used to indicate the panels in the text. Furthermore, the authors must standardize in the whole work the way of indicating the panels according to the template. Please, correct.

In Figure 7 the measurement scale is still invisible.

The way of presenting the peaks lists of spectrophotometric analyzes is not acceptable. Please consult  https://doi.org/10.3390/nano12020233  or https://doi.org/10.3390/nano11102662 to take an example and correct accordingly.

The experimental data are still too poor. Please, consider the works proposed above to improve.

In  13C and 1H NMR 13 and 1 must be superscript. Please, correct.

There are still parts in bold where bold is not required. Please, correct.

The authors say they cannot determine the molecular weight of the polymer because they are not using GPC. There are other ways. Since they have determine the viscosity of polymer they could exploit it to determine its  Mn.

The authors should provide the graphs of the mathematical models of which they have provided only the coefficients of determination.

These are only a part of the issues that go on hampering the publication of this paper.

Author Response

Dear Reviewer 

We would like to thank you for the valuable comments. We responded to all the comments. Attached please find the response file addressing the reviewers' comments point-by-point.  

Reviewer 2 Report

The reviewer recognizes the affords for improvement of the manuscript. The authors responded and corrected most of comments. However, they refused to evaluate the bioactivity on Covid-19. Thus, the authors must remove this information from the title. Importantly, the kinetic calculation was performed to confirm release mechanism. The data is suitable but the author must provide histograms for these determination of values “R2”, constant and “n” in a supporting information file. After the corrections, the manuscript can be accepted.

Author Response

Dear Reviewer 

We would like to thank you for the valuable comments. We responded to all the comments. Attached please find the response file addressing the reviewers comments point-by-point.  

Round 3

Reviewer 1 Report

I have reviewed the manuscript by Dahmash and co-workers for the third time, and I am pleased to find that the authors have revised the previous version satisfying practically all my requests.

The work is now complete and in the format requested by Polymers.

I congratulate the authors for the great work they have done.

Unfortunately, there are still two errors to be corrected, and one is a gross error of concept. The authors in the peaks lists of NMR spectra wrote that the proton spectra were acquired at 500 MHz (and that's okay) but then claim that the carbon spectra were also acquired at 500 MHz, which is impossible.

The MHz of 13C NMR is about a quarter of that of 1H NMR, so either the authors have a 2000 NMR (which does not exist on the face of the earth) or they need to correct such data.

Furthermore, in the description of the instruments, the authors state that they also used a 300 MHz Bruker NMR instrument, which in reality is never used. This is curious and makes me think about, but I don't want to investigate further.

Author Response

Dear reviewer 

Attached please find the corrected manuscript and the letter that outlines the changes that were done as per your kind recommendations. 

Reviewer 2 Report

It is acceptable.

Author Response

Dear Reviewer 

Many thanks for your valuable recommendations.